# Tuning Monocytes and Macrophages for Personalized Therapy and Diagnostic Challenge in Rheumatoid Arthritis

**DOI:** 10.3390/cells10081860

**Published:** 2021-07-22

**Authors:** Leszek Roszkowski, Marzena Ciechomska

**Affiliations:** Department of Rheumatology, National Institute of Geriatrics Rheumatology and Rehabilitation, 02-635 Warsaw, Poland; leszekroszkowski@poczta.fm

**Keywords:** rheumatoid arthritis, monocytes, macrophages, personalized therapy, biomarkers, epigenetics, small molecules, biologics, heterogeneity, single-cell sequencing

## Abstract

Monocytes/macrophages play a central role in chronic inflammatory disorders, including rheumatoid arthritis (RA). Activation of these cells results in the production of various mediators responsible for inflammation and RA pathogenesis. On the other hand, the depletion of macrophages using specific antibodies or chemical agents can prevent their synovial tissue infiltration and subsequently attenuates inflammation. Their plasticity is a major feature that helps the switch from a pro-inflammatory phenotype (M1) to an anti-inflammatory state (M2). Therefore, understanding the precise strategy targeting pro-inflammatory monocytes/macrophages should be a powerful way of inhibiting chronic inflammation and bone erosion. In this review, we demonstrate potential consequences of different epigenetic regulations on inflammatory cytokines production by monocytes. In addition, we present unique profiles of monocytes/macrophages contributing to identification of new biomarkers of disease activity or predicting treatment response in RA. We also outline novel approaches of tuning monocytes/macrophages by biologic drugs, small molecules or by other therapeutic modalities to reduce arthritis. Finally, the importance of cellular heterogeneity of monocytes/macrophages is highlighted by single-cell technologies, which leads to the design of cell-specific therapeutic protocols for personalized medicine in RA in the future.

## 1. Introduction

Rheumatoid arthritis (RA) is a chronic inflammatory autoimmune disease affecting approximately 1% of the world’s population, and is influenced by multiple genetic, epigenetic and environmental factors. RA is a systemic disease characterized primarily by chronic arthritis that ultimately leads to joint damage and loss of function, followed by a subsequent deterioration in the patient’s physical and social functions [1]. Across Europe, rheumatic and musculoskeletal diseases (RMDs), of which RA is the most common inflammatory arthritis, represent a huge economic burden with an upward trend driven by demographic and behavioral changes. These diseases are the leading cause of disability and premature retirement among workers because they cause more functional limitations in adults than any other group of disorders. Therefore, early detection as well as cell-specific and personalized treatments are crucial in RA management [2].

Circulating monocytes strongly contribute to RA progression due to production of pro-inflammatory molecules and infiltration of inflamed synovium where they differentiate into macrophages [3]. Indeed, activated monocytes/macrophages produce pro-inflammatory cytokines (TNF-α, IL-1β, GM-CSF and IL-6), metalloproteinases (MMP-3 and MMP-12) and chemotactic agents (CCL2, CCL3, CCL5, CX3CL1 and IL-8) [4,5]. These cells also produce pro-inflammatory mediators which have been recently explored as novel diagnostic biomarkers of RA activity, including serum amyloid A (SAA), toll-like receptors (TLRs) and their ligands, and high mobility group box 1 (HMGB1). Importantly, radiological progression of joint destruction correlates with macrophage synovial infiltration. Macrophage depletion by antibodies or chemicals agents reduces synovitis, suggesting a critical role of monocytes/macrophages in the pathogenesis of RA, both in vitro in humans and in vivo in animal models [6,7].

Historically, macrophages have been classified according to a linear scale, with M1 macrophages representing one extreme and M2 macrophages representing the other. Polarization of macrophages is plastic and reversible. M1 polarization occurs in the early stages of the inflammatory response, whereas resolution of inflammation, vasculogenesis and tissue remodeling are dominated by M2 polarization. The sequential occurrence of two polarization states is an absolute prerequisite to the correct termination of the inflammatory response and the repair of normal tissue after injury. Changes in shifts between macrophage polarization states result in chronic pathologies of inflammatory and autoimmune diseases such as RA. Therefore, to assess disease activity and effectiveness of treatment, the M1/M2 ratio is required instead of changing a single M1 or M2 [8,9]. It has been demonstrated that patients with active RA show an increased M1/M2 ratio which promotes osteoclastogenesis, and in patients with remission or low disease activity they show a phenotype similar to M2 [10]. New insights into the use of single-cell RNA sequencing (scRNA-seq) technologies could help to uncover cellular heterogeneity by identification of novel subpopulation of monocytes/macrophages in RA (more in Section 6). Furthermore, distribution of circulating monocytes is also important in the prediction of clinical response to anti-TNF or methotrexate (MTX) in RA patients [11,12]. Indeed, RA monocytes can be further differentiated not only into macrophages but also into osteoclasts which are directly involved in bones and joint destruction [13] (Figure 1). Therefore, osteoclast can be used as biomarkers of disease severity as well as predictors of response to disease-modifying antirheumatic drug (DMARD) therapy [14]. Monocytes can also differentiate into CD1c+ inflammatory DC (infDCs) under the influence of GM-CSF [15]. Synovial CD4^+^ T-cell-derived GM-CSF supports the differentiation of an infDCs population in RA. On the other hand, upon specific stimulation, including dexamethasone and vitamin D3 or IL-10 treatment, circulating monocytes can further differentiate into tolerogenic dendritic cells (tolDC). TolDC are involved in the induction of self-tolerance and long-term remission while leaving protective immunity intact [16]. Overall, these studies indicate that monocytes can either differentiate into pathogenic pro-inflammatory M1 macrophages and osteoclast or anti-inflammatory M2 macrophages and tolDC, suggesting that cell-mediated therapies and unique inflammatory biomarkers produced by monocytes/macrophages provide a more tailored treatment and diagnostic strategy in RA. Recently, precision medicine has gained an important role in the treatment of RA, thus immunomodulatory function of monocytes/macrophages could be a novel tool in precision medicine focusing on individual RA patients.

## 2. Role of Macrophages in RA Synovium

As circulating precursors of macrophages, monocytes have different origins including the embryonic yolk sac, postpartum bone marrow and fetal liver, and can be divided into three subgroups [17,18]. Based on the expression of CD14 and CD16 (surface molecules), monocytes are divided into classical (CD14^++^CD16-), intermediate (CD14^++^CD16^+^), and non-classical (CD14^+^CD16^++^) subgroups [19]. Classical monocytes are the most important subgroup of circulating monocytes and function as phagocytic scavenger cells. In response to lipopolysaccharide (LPS) and activation of the immune complex, they produce high levels of TNF-α, IL-6, IL-10 and IL-1β [20]. Studies have shown that monocytes, which can differentiate into osteoclasts and cause joint damage, have a classical phenotype. These classical monocytes are also precursors of circulating osteoclasts and the expression of the RANK receptor on their surface is increased [21]. Intermediate subsets of monocytes produce high levels of pro-inflammatory cytokines IL-1β, TNF-α, IL-6 in the synovium in patients with RA. Subsequently, these monocytes are able to differentiate into inflammatory macrophages (M1) which promote synovitis as well as activate and promote Th17 cells [22,23]. Due to their properties of immediate extravasation from the vascular endothelium during tissue damage, non-classical monocytes are responsible for an early inflammatory response. They also differentiate into resident macrophages (M2) and are later involved in inflammation resolution [24]. Interestingly, using human phenotypic analyses, Ambarus et al. revealed that there is no difference in monocyte subgroups distributions (classical, intermediate and non-classical) between people with RA and healthy donors [25].

The synovium and its infiltrating macrophages play an extremely important role in the formation, development and maintenance of arthritis in RA (Figure 2). Normal synovium consists of two separate parts: the lining layer of the synovial membrane and synovial sublining layer. The inner layer of the lining consists of fibroblast-like synoviocytes (FLS) and macrophage-like synovial cells (MLS). The synovial sublining layer is made up of synovial macrophages (MS), blood vessels and other cells [26]. Analysis of the FLS and MLS transcriptome isolated from RA patients confirmed that MLS were macrophages with a strong tendency to inflammation [27]. Research suggests that MLS originate from the bone marrow, possibly due to their resemblance to mononuclear phagocytes elsewhere [28]. Another study found that macrophage populations in the synovial lining are embryo-derived, however more research is needed to verify this hypothesis [29]. Interestingly, immunohistological analysis of frozen sections of pannus tissue from synovium of RA patients showed a difference in the phenotypes of synovial lining macrophages and macrophages at the cartilage junction [30]. Synovial macrophages may not only play an important role in the inflammatory process but may also be involved in angiogenic processes [31]. In addition, accumulation of pro-inflammatory S100A8/A9 proteins (also known as myeloid related proteins 8/14 or calprotectin) produced by monocytes/macrophages was observed in the synovium of RA patients in the sublining layer. Mature macrophages positive for late inflammatory macrophage markers were more abundant in the inner layers of the lining, indicating the difference between infiltration (sublining) and phenotypes of tissue-resident macrophages (lining) [32]. All these data indicate that macrophages are directly involved in synovitis resulting in joint destruction and RA progression.

## 3. Genetic, Epigenetic and Environmental Factors Modulating Monocytes/Macrophages in RA

Based on genome-wide association GWAS studies, genetic factors like HLA-DRB1 shared epitope, PTPN22 and MCP-1 risk alleles play an important role in RA pathogenesis [33,34]. Of note, monocytes are the primary source of MCP-1 among PBMCs and have upregulated expression of HLA-DRB1, suggesting a strong link between genetic association and involvement of monocytes in RA susceptibility [35,36]. Furthermore, recent discoveries in microbiome research have clearly demonstrated the contribution of environmental factors, including chronic microbial infections, to susceptibility to RA. Indeed, bacterial nucleic acids were present in the joints of RA patients [37]. Transcriptomic analysis of RA synovial tissue revealed activation patterns of infiltrating monocytes/macrophages. These patterns have corresponded to activation induced by microbial stimuli, whereas top proteins including MCP-1 and S100A8/A9 secreted by these cells in RA synovial tissue correlated with disease activity and reflected RA synovitis in blood [38]. Many research studies suggested that genetic contributions to RA disease susceptibility are not sufficient to completely explain the disease. The concordance rate of RA for monozygotic twins is only 15%, implying that inherited DNA sequences along with epigenetic factors contribute to RA development [39].

Epigenetics are defined as reversible and heritable modifications in gene function, however these modifications do not alter the primary DNA sequence but instead affect how cells activate or suppress specific genes in response to environmental stimuli. Three main epigenetic mechanisms have been described including DNA methylation, non-coding RNA species and histone modifications. All these changes contribute to the breakdown of immune tolerance and to chronic inflammation driven by immune cells, including monocytes/macrophages in RA [40]. Therefore epigenetic-based therapeutics that control autoimmunity and pathogenic role of monocyte/macrophage subsets in RA have broad implications for the pathogenesis, diagnosis and management of rheumatic diseases.

### 3.1. Role of DNA Methylation

DNA methylation is a process by which methyl groups are added to the DNA molecule and is induced by a highly conserved family of DNA methyltransferases (DNMTs). DNA hypermethylation results in gene silencing. Many studies have revealed that the DNA methylation pattern is impaired in RA, affecting immune cells including monocytes and consequently influencing immune responses [41]. Using global DNA methylation profiling, it has been demonstrated that monocytes from RA patients display a wide range of changes in their DNA methylome and such changes correlate with DAS28 [42]. Enrichment analysis of differentially methylated CpG revealed that HLA-DPB2, ETS1 and FOXO3 genes were hypomethylated, whereas CREBBP, SOCS7 and TRAF genes were hypermethylated in RA monocytes. These data suggest a direct link between altered methylome in RA monocyte resulting in activation of inflammation-associated cytokines and disease activity. Microarray data also identified 193 aberrantly methylated genes in RA monocytes. Further enrichment analysis demonstrated elevated methylation of SIRT1, SKP2, TUBA1A, IMP3, EXOSC5 and SMAD4 genes and reduced methylation of KRAS in samples from RA patients compared to healthy individuals [43]. In addition, hypomethylation of the CYP2E1 and DUSP22 promoters in RA monocytes correlates with RA activity [44]. CYP2E and DUSP22 are involved in pro-inflammatory cytokines production. Interestingly, using genome-wide DNA methylation profiles, it has been demonstrated that methylation profile of monocytes isolated from adults were different compared to the methylation profile of monocytes from children and were correlated with increased production of IL-8, IL-10, and IL-12p70 in response to TLR-4 and TLR-2 stimulation [45]. These data suggest that age-related DNA methylation of monocytes mediates pro-inflammatory cytokines production. Recently, machine learning models generated from transcriptome data of monocytes and DNA methylome data of PBMCs have demonstrated 80.3% accuracy in the prediction rate of RA patients’ response to adalimumab, paving the path towards personalized TNF-inhibitor (TNF-i) treatment strategies [46]. Furthermore, significantly increased expression of ten-eleven translocation1 (TET1) genes was observed in RA monocytes. TET1 enzymes are involved in the DNA demethylation process. Interestingly, treatment with MTX restored DNA hypomethylation status of RA monocytes. These monocytes were no longer different from HC monocytes. Overall, the DNA methylation signature of monocytes could be an important feature for providing a better RA subtype identification in order to accelerate development of precision medicine.

### 3.2. Role of miRNA

miRNA can function both by promoting inflammatory phenotypes and by acting through negative feedback loops to limit inflammation [47]. Compelling evidence supports that miRNAs serve as influential mediators of monocyte-driven inflammation. For example, miR-155, which was increased in RA monocytes, promoted pro-inflammatory M1 phenotypes and suppressed M2 features via increased production of CCL3, CCL4, CCL5 and CCL8 and correlated with DAS28-ESR [48,49]. In addition, using global microarray analysis, miRNA-125a was identified as a highly upregulated biomarker in monocytes from children with active juvenile idiopathic arthritis (JIA) and correlated with systemic features of the disease [50]. Overexpression of miRNA-125a in THP-1 cells promoted a monocyte phenotype similar to the one seen in systemic JIA. Ren et al. identified that monocytes from RA patients were resistant to spontaneous apoptosis compared to HC monocytes due to increased expression of miRNA-29b [51]. Such upregulation of miR-29b expression was also correlated with CRP, RF and ESR in RA patients. miRNA-29b negatively regulates pro-inflammatory marker HMGB1 via a direct bind to the 3’-UTR of HMGB1. Importantly, exogenous delivery of miRNA-29b resulted in upregulation of pro-inflammatory IL-1α/β, TNF-α, IL-6, IFN-α, IL-8 and IL-15 in monocytes. Ammari et al. demonstrated that inflammatory monocytes had reduced expression of miRNA-146a in RA patients [52]. Other study indicated that miR-146a deficiency enhanced inflammatory arthritis severity and bone erosion in a mouse model [52], while exogenous delivery of miRNA-146a into monocytes resulted in suppression of pathogenic bone erosion during inflammatory arthritis. Using a high throughput approach, our previous results revealed a global miRNA dysregulation and monocyte transcriptome in patients with autoimmune rheumatic diseases. Indeed, based on next generation sequencing (NGS) analysis, we identified miRNA-RNA pairs in monocytes which are important in RA pathogenesis [53]. Further validation predicted that miRNA-146b negatively regulated anti-inflammatory RARA in RA monocytes and correlated with clinical parameters including DAS28. In addition, ROC analysis revealed that enhanced expression of circulating miRNA-146b in sera and synovial fluids was a better disease activity biomarker than the CRP level. Similarly, miRNA-mRNA co-sequencing and functional analysis identified miRNA-26a as a new candidate which is predicted to negatively regulate IFN-regulated genes in systemic sclerosis (SSc) monocytes [54]. This implies that reduced expression of miRNA-26a may be involved in the pathogenic IFN signature in SSc monocytes. Given the observed correlations between altered expression of miRNAs in monocytes, their miRNA-regulated transcripts and disease activity, these data provide strong evidence that therapeutic delivery of specific miRNA to monocytes may represent a possible treatment strategy for pathogenic arthritis or other rheumatic diseases. In addition, unique dysregulation in miRNA expression in monocytes could be used as a robust biomarker for RA risk assessment, disease progression and therapy response. Therefore, we expect that future advancements using miRNA will focus on more monocytes-specific personalized approach.

### 3.3. Role of Histone Modification

Histone modifications are covalent modifications to histone proteins that facilitate or block DNA accessibility. These post-translational modifications in histone tail domains and core domains comprise of different processes including methylation, acetylation, citrullination, phosphorylation, ubiquitination, ribosylation, sumoylation, biotinylation or glycosylation. It is known that dysregulation in histone modifications contributes to RA development [55]. Therefore, small molecule inhibitors blocking histone modification may represent a new therapeutic approach in RA. For instance, inhibition bromodomain and extra-terminal (BET) family proteins by chemical agent I-BET151 suppressed the expression of IFN-stimulated gene (ISG) including IFIT1, IFIT2, and IRF1 and production of chemokines (CXCL10, CXCL11) in LPS or TNF-α stimulated monocytes [56]. BET proteins control the transcription of various pro-inflammatory genes by interacting with acetylated histones (mainly histones H3 and H4) in RA, therefore inhibition of BET proteins results in pro-inflammatory cytokines reduction [57,58]. In addition, silencing of BET proteins by small interfering RNA reduced TNF-α expression in HC and RA monocytes-derived macrophages [59]. Cribbs et al. have also showed that monocytes reduced TNF-α production upon co-culture with NK cells stimulated with BET inhibitor JQ1 [60]. Similarly, the immune complex-mediated activation of monocytes was reduced by JQ1 in the lupus model with MRL-lpr mice [61]. Histone Deacetylase inhibitor (HDAC-i) trichostatin A (TSA) induced apoptosis in RA synovial macrophages and suppressed IL-6 and IL-8 production in synovial biopsies. However, due to TSA high cytotoxic effects, this drug was only tested in preclinical studies [62]. Our previous results showed that monocytes treated with small molecules inhibitors including DZNep (histone methyltransferase inhibitor), but not apicidine (histone deacetylase inhibitor), increased production of pro-fibrotic collagen type 1, TIMP-1, IL-8 and enhanced SSc development [63,64]. Elevated levels of citrullinated H3 in monocytes from ACPA-positive subjects strongly correlated with increased risk of RA development, suggesting the presence of a unique citrullination signature in monocytes during the early stage of RA [65]. In addition, TLR stimulation of monocytes induced pro-inflammatory cytokine production, including TNF-α, MIP1-β, IFN-α and PAD4 enzymes. PAD4 enzyme is involved in the citrullination process that generates citrullinated autoantigens in RA. Interestingly, etanercept and adalimumab treatment downregulated acetylation of H3 and H4 in the CCL2 gene promoter and subsequently CCL2 synthesis in RA monocytes [66]. These data suggest that anti-TNF therapy modifing histones result in pro-inflammatory CCL2 inhibition in RA monocytes. Therefore, identification of the unique profile of histone modifications in monocytes and its regulation mechanism by small molecules may find broad implications for the pathogenesis, diagnosis and management of rheumatic diseases.

## 4. Targeting Monocyte/Macrophage Biomarkers in RA Diagnostics

Monocytes/macrophages express many specific molecules which play a role in RA pathogenesis i.e.: TNF-α, IL-1β, S100A8, S100A9, GM-CSF, IL-6, HMGB, SAA, MMP-3, MMP-12, MCP-1/CCL2, CCL3, CCL5, CX3CL1 and IL-8. Each of them can be considered a biomarker associated with RA. For the purpose of this chapter, we focused on novel biomarkers which could be helpful in RA diagnosis [67,68]. On the other hand, inhibition or neutralization of specific molecules excessively produced by monocytes/macrophages may find a broad implication in RA therapy (Figure 3).

### 4.1. Targeting Toll-Like Receptors (TLRs)

Innate immune cells, such as monocytes/macrophages, neutrophils and dendritic cells, express both cell surface and intracellular TLRs. TLRs recognize microbial products, cellular debris from necrotic cells and inflammatory tissues such as lipoteichoic acid (LTA), peptidoglycan, fibronectin products, LPS, endogenous Hsp60 molecules and hyaluronic acid [69]. TLRs play a key role in the nonspecific response to both infections (most recent research has been done on COVID-19) and immune diseases [70]. The role of TLRs in RA is very important through their increased expression in blood circulation and infiltrating cells in an inflamed joint. In addition, high TLR-2 expression in monocytes and CD16+ synovial macrophages of RA patients induced production of increased amounts of TNF-α, IL-6 and IL-8 [71]. The research proved that the subgroups of classical and intermediate monocytes in the peripheral blood and synovium of RA patients showed a significantly elevated level of TLR-2. All three subgroups of monocytes also showed elevated levels of TLR-9. The triggering pathways of TLR-2 and TLR-9 in these monocyte subgroups elevated the production of the pro-inflammatory cytokines IL-1β, IL-6, TNF-α and the chemokine MIP-1 [72]. Importantly, anti-citrullinated protein antibodies (ACPA) and rheumatoid factor (RF) can activate macrophages by binding to TLRs, binding to membrane citrullinated vimentin, and then causing bone loss in RA patients [73]. ACPA has also been shown to induce TNF-α production by macrophages through conjugation with TLRs, Fcγ receptors and the surface expressed citrullinated protein Grp78 [74,75]. Studies demonstrated that synovial fibroblasts express increased amounts of TLR-3, TLR-4 and TLR-7/8 associated with persistent arthritis and joint destruction in RA patients [76,77]. The discovery of TLRs and their functions opened new diagnostic and therapeutic perspectives in autoimmune diseases, especially in RA. Blocking TLR signals is an attractive therapeutic approach, as demonstrated with an anti-TLR-2mAb capable of reducing the spontaneous release of pro-inflammatory cytokines from RA synovial tissue explant cultures [78]. Recent studies have found that human anti-TLR-4 IgG2 inhibited expression of TNF-α, INF-β and IL-6 genes in murine peritoneal macrophages [79]. Another molecule used in animal studies is TAK-242. TAK-242 is a small molecule which acts by inhibiting the TLR-4-mediated signaling and by suppressing the release of inflammatory cytokines [80]. Moreover, NI-0101 is the first humanized monoclonal antibody to block TLR-4. Unfortunately, a Phase II study showed that blocking the TLR-4 pathway alone did not improve disease parameters [81].

### 4.2. Targeting Calprotectin (CLP)/S100A8/A9

A very interesting new diagnostic protein used in RA is CLP, a member of the S100 protein family, which is a heterodimeric complex of S100A8 and S100A9. CLP expression was found primarily in granulocytes and monocytes and during early stages of macrophage differentiation [82]. Research showed that CLP in monocytes was actively released by an intact network of microtubules upon activation of protein kinase C [83]. Importantly, S100A8 is responsible for CLP complex activity, while S100A9 protects S100A8 from degradation [84]. It has been proven that CLP induces the production of TNF-α, IL-6, IL-1β and IL-8 in monocytes, playing an important role in the induction and maintenance of inflammation [82]. In recent years, all studies showed that CLP levels in the blood and synovial fluid were significantly elevated in RA patients, and that levels correlated positively with disease activity [85]. Moreover, CLP levels are also closely correlated with radiographic progression of joint damage, ultrasound-determined synovitis and therapeutic response in RA patients [86,87,88]. Interestingly, studies have found that CLP was more strongly associated with ultrasound-detected synovitis and clinical outcomes of inflammation than CRP, ESR, IL-6, S100A12 and VEGF [89]. Furthermore, it has been suggested that CLP may be a valuable supplement to conventional inflammatory markers in the evaluation of RA patients during the follow-up period after bDMARD treatment [89]. All these reports suggest that CLP can be considered as an effective biomarker used in precision medicine. However, it should be remembered that its increase was also observed in other inflammations associated with autoimmune, cancer and infectious diseases (including COVID-19) [90,91]. New drugs are currently being investigated in the early stages that would affect the CLP-mediated inflation by blocking secretion, inhibiting CLP expression, interacting with the CLP receptor and inducing innate immune tolerance [92].

### 4.3. Targeting High Mobility Group Box-1 (HMGB1) Protein

Another important molecule that is largely released from stimulated monocytes/macrophages is HMGB1. It binds to chromosomal DNA, but also to TLR-3, TLR-4 and the receptor for advanced glycation end products (RAGE), which activates the nuclear factor -B (NF-κB). Activation of NF–κB mediates the upregulation of leukocyte adhesion molecules, as well as the production of pro-inflammatory cytokines and angiogenic factors that promote inflammation [93]. HMGB1 has been studied extensively because it has a clear relationship with insulin resistance and diabetes, obesity and polycystic ovarian syndrome, as well as immunological conditions such as systemic lupus erythematosus (SLE), RA and infectious diseases, including in COVID-19 [94,95,96,97,98,99]. Serum concentration of HMGB1 and its nucleosome complex has been found to be higher in RA patients compared to healthy controls and contributes to disease progression. Many studies focusing on the relationship between RA and HMGB1 have shown that HMGB1 plays a significant role as a pro-inflammatory cytokine in the pathogenesis and development of RA [100,101]. Studies also showed that the serum concentration of HMGB1 in RA patients was correlated with the disease activity score DAS-28 [102]. Increased concentration of HMGB1 in RA patients was found not only in the serum, but also in the synovial fluid. In synovitis, HMGB1 expression was particularly present in macrophages in vascular endothelial cells [103]. Importantly, upon exposure to HMGB1, macrophages in the synovial fluid promote increased expression of RAGE, TLR-4 and TLR-2 and may be activated to mediate the release of pro-inflammatory cytokines [104]. This important role of HMGB1 in the course of RA is considered not only a potential marker of disease activity, but also a therapeutic target. Currently, therapeutic interventions targeting HMGB1 are being made, such as anti-HMGB1 antibodies, anti-RAGE antibodies, etc. For now, all studies are in preclinical models and further results are expected [105].

### 4.4. Targeting Serum Amyloid A (SAA)

SAA is a family of proteins produced by macrophages, among others, that is an acute phase biomarker in many inflammatory rheumatic diseases. In a study by Jumeau et al., they confirmed the induction of SAA1 and SAA2 gene transcription in monocytes and macrophages derived from monocytes in response to combined lipopolysaccharide and glucocorticoid stimulation [106]. In addition, SAA proteins were detected by immunohistochemistry (IHC) in RA macrophages [107]. The best known diseases for which SSA may be useful are RA, JIA, ankylosing spondylitis, various types of vasculitis, sarcoidosis, psoriatic arthritis, systemic sclerosis, SLE, familial Mediterranean fever, and, based on recent work, COVID-19 [108,109,110,111,112,113,114,115,116,117,118,119]. It is very important that SAA can be used as a diagnostic marker for RA as its level is elevated compared to osteoarthritis patients not only in serum but also in synovial fluid as a result of local production [120]. Many studies have shown a significant correlation between serum SAA concentration and RA disease activity [121]. It is surprising that SAA is a more sensitive marker of disease activity in RA than the much more commonly used CRP or ESR, which is important and even useful during pregnancy and for the detection of subclinical inflammation in patients with CRP within the reference range [122,123,124]. SAA is also important for monitoring RA disease activity in patients receiving anti-TNF therapy because this therapy reduced CRP even without reducing disease activity, while SAA was not affected [125]. Another study assessed that SAA is also a sensitive biomarker of the response to tofacitinib (Janus kinase inhibitor) in patients with active RA [126]. It has also been found that SAA can predict swollen joint count (SJC28), tender joint count (TJC28) or patient global assessment (PGA), so SAA has been classified as multi-biomarker disease activity (MBDA) for RA [127]. In addition, researchers investigated that SAA baseline levels could be used to predict remission over 12 months in RA patients [128]. It is extremely important that continuously elevated SAA levels posed a risk of developing amyloidosis, therefore serum SAA measurements are useful in identifying patients in need of more intensive treatment, especially biological and targeted synthetic DMARDs [129]. In another study, anti-SAA as well as anti-SAA1α autoantibodies were detected in intravenous immunoglobulin and were shown to reduce levels of IL-6 protein released from SAA/SAA1α treated monocytes. This suggested that naturally occurring anti-SAA and anti-SAA1α antibodies may play a physiological role in lowering pro-inflammatory cytokines and may be an important therapeutic option in RA patients [130].

## 5. Modulation of Monocytes/Macrophage in RA Therapies

The main goals of RA treatment based on EULAR and ACR recommendations are: reducing joint pain, joint swelling, alleviating synovial inflammation, preventing damage to joints and articular structures, inhibiting disease progression, preventing bone erosion and restoring joint mobility. To achieve these objectives, various medications are routinely prescribed to RA patients. Some agents work by reducing the symptoms of RA (pain, inflammation and joint stiffness), some medications slow or stop the progression of the disease and prevent damage bone and cartilage [131]. Overall, there are five major classes of drugs that are currently in use: (1) non-steroidal anti-inflammatory drugs (NSAIDs), (2) corticosteroids, (3) conventional synthetic DMARDs, (4) targeted synthetic DMARDs and (5) biological DMARDs [131].

Knowing that macrophages play a key role in cytokine-induced inflammation in RA, targeting monocytes/macrophages and their mediators is a widely used new approach to treating RA patients. Understanding the complex network of disease-involved cytokines that are produced by macrophages has paved the way for tremendous therapeutic advances targeting a variety of factors such as TNF-α, IL-1, IL-6, IL-8, CCL2, MMP-3, MMP-12 and GM-CSF [4,5]. In this chapter, we focus on modern, personalized therapies that have a macrophage/monocyte-mediated linkage.

### 5.1. Modulation of Monocytes/Macrophage by Biologics

In the treatment of RA patients, the most commonly used biological treatment is inhibition of TNF with antibodies or soluble receptors. Currently there are five available drugs targeting the TNF pathway in RA treatment: infliximab, adalimumab, etanercept, golimumab and certolizumab pegol. It is very important that TNF produced by macrophages has a positive effect on their own survival, maintaining their presence in the synovial fluid and peripheral blood with inflammation (Table 1) [132]. Post-treatment studies with anti-TNF investigated the tissue kinetics of monocytes in RA and revealed the key function of TNF [133]. Anti-TNF therapy can increase TGF-β levels, which in turn stimulates regulatory T cells to suppress macrophages [134]. These extremely important drugs reduce the number of infiltrating granulocytes and macrophages in the RA synovium, as well as the expression of IL-8, MCP-1, IL-1, IL-6, GM-CSF and chemokines [135]. Etanercept and adalimumab induced polarization of macrophages in the anti-inflammatory direction due to activation of surface markers and expression of cytokines characterizing M2 [136]. All of this results in a reduction in arthritis, pain and fatigue in RA patients. TNF inhibitors are recommended as a second line of treatment, preferably in dual therapy with MTX. It is worth mentioning that MTX impairs monocyte chemotaxis, reduces their survival, and also inhibits the recruitment of monocytes to inflammatory tissue, especially joints [137]. Moreover, glucocorticoids, MTX and leflunomide work by shifting the macrophages of RA patients to an M2-like state [138].

Another target associated with macrophages in the treatment of RA is IL-6. Numerous studies have demonstrated the key role of the pleiotropic cytokine IL-6 in RA and the very positive effects of its blocking [139]. In RA, there are six drugs that focus on binding to IL-6: tocilizumab (the first anti-IL-6 receptor antibody), sarilumab, olokizumab, clazakizumab, vobarilizumab and sirukumab. Among its many effects, the most widely used tocilizumab affects macrophage function by increasing the Peroxisome Proliferator-Activated Receptor (PPARγ). Moreover, tocilizumab appears to be more effective in shifting the macrophages towards an anti-inflammatory phenotype compared to abatacept and etanercept [140]. Studies also showed that tocilizumab induced monocyte apoptosis by interacting with Fc receptors and inhibiting IL-6 mRNA expression in monocytes [141].

Different drugs used in RA are IL-1 inhibitors, mainly anakinra (an IL-1 receptor antagonist), canakinumab and rilonacept, but due to their lower efficacy than other biological drugs, they are used less frequently [142]. Another promising group of drugs that are currently in clinical trials are neutralizing antibodies against granulocyte-monocyte colony-stimulating factor (GM-CSF). GM-CSF is a pro-inflammatory cytokine that can induce the differentiation of monocytes to M1 at sites of RA inflammation. The most famous of these drugs are mavrilimumab (monoclonal IgG4 antibody against the human alpha chain of GM-CSF receptor) and otilimab (formerly MOR103—monoclonal recombinant IgG1 antibody with a high affinity against GM-CSF), and their efficacy and safety were confirmed in phase II studies in patients with RA. The other drugs are gimsilumab, namilumab and lenzilumab [143,144]. In CIA mice, treatment with an anti-GM-CSF receptor antibody resulted in reduced clinical markers of arthritis and synovial macrophages [145]. Another biologic drug modulating pro-inflammatory functions of monocytes/macrophages and T lymphocytes is abatacept. Abatacept is a CTLA4-IgG1 fusion protein that binds to CD80 and CD86 on antigen presenting cells (APCs) and blocks interaction with CD28 on T cells, thereby inhibiting T cell activation [146]. Abatacept also inhibits the osteoclastogenesis of human peripheral monocytes, thus inhibiting bone resorption and preserving bone mass [147]. Abatacept has also been found to modulate the production of pro-inflammatory cytokines (TNF-α, interleukine IL-6, IL-1β) [148]. Drugs that target other cytokines produced by monocytes/macrophages, including IL-7, IL-12, IL-15, IL-18, IL-21, IL-23, IL-32 and IL-33, are currently in clinical trials. In particular, anti-IL-12 (ABT-874), anti-IL-15 (HuMax-IL15), anti-IL-18, ustekinumab (human monoclonal anti-IL-12/23 p40 antibody) and guselkumab (human monoclonal anti-IL-23 antibody) have shown promising results in murine models of arthritis and other rheumatoid diseases which are currently under further investigation [149,150,151,152].

### 5.2. Modulation of Monocytes/Macrophages by Small Molecules

Small molecules play an increasing role in the treatment of RA [153]. JAK inhibitors (JAK-i) inhibit the JAK-STAT pathway activity, and since many cytokine receptors, including IL-6, IFN type 1, IL-1, IL-17 and GM-CSF, signal through JAKs, they can be used to block cytokine signaling which is beneficial in the treatment of RA [154]. The three approved drugs for the treatment of RA JAK-i are: tofacitinib (JAK1/JAK3-i), baricitinib (JAK1/JAK2-i) and upadacitinib (JAK1-i). We now know that their function is to block the inflammatory factors secreted by macrophages, but additional studies are needed to assess their full role in macrophage function [155]. A new drug that has been successfully investigated in RA patients in a phase II trial is spebrutinib (CC-292), the first irreversible Bruton’s tyrosine kinase (BTK) inhibitor [156] and fenebrutinib, a highly selective non-covalent BTK inhibitor [156]. The mutations in the BTK genes have been shown to be associated with X-linked immunodeficiency in mice, and the macrophages of these mice produce less TNF-α, nitric oxide and IL-Iβ [157]. Many other signaling pathways in RA have been studied in both human and animal models, including the serine-threonine kinases SIK2 and SIK3, MAPK/ERK kinases (MEK), spleen tyrosine kinases (Syk) and p38 MAPKs. Unfortunately, they are not yet clinically successful to this day, as demonstrated by the challenge of selecting key signaling targets from kinases-mediated complex networks [158,159,160,161].

### 5.3. Modulation of Monocytes/Macrophages by Other Therapeutic Modalities

Research is also ongoing into a specific macrophage strategy for RA, which may depend on related transcription factors, metabolites and factors responsible for macrophage death. Attempts are being made to alter the phenotype of pro-inflammatory monocytes and macrophages, or to reduce resistance to death of inflammatory macrophages. Some recently discovered interleukins may also indirectly regulate macrophage function. One of them, IL-34, has been shown to promote monocyte survival, proliferation and differentiation into macrophages [162]. In contrast, IL-35 promotes the FLS apoptosis induced by TNF-α and stimulates M2 polarization of macrophages, thereby inhibiting inflammation in mouse CIA [163]. In addition, IL-37, which indicates M2 polarization of macrophages and IL-38, induced anti-inflammatory effects in animal models of arthritis and THP-1 in vitro [164,165].

Macrophage synovial hyperplasia expresses CD64 at a high level, and the CD64-targeting immunotoxin promotes their selective elimination through apoptotic cell death [166]. Therefore, new human cytolytic H22 (scFv)-MAP fusion proteins have been developed that induce apoptosis of CD64^+^ M1 macrophages in murine arthritis models by converting M1 into M2 macrophages, exerting anti-inflammatory effects [167].

An interesting new method of treating RA is macrophage-derived microvesicle-coated nanoparticles (MNPs). This treatment is inspired by the inner capacity of the macrophages to control the inflammation. The model drug, tacrolimus, was encapsulated into MNPs and significantly inhibited the progression of RA in CIA mice by targeting Mac-1 and CD44 [168]. CD40 is another costimulatory molecule found on macrophages and is currently being intensively studied. Both the neutralizing anti-CD40 monoclonal antibodies and the non-antibody scaffold protein VIB4920 (a human CD40L inhibitor) have been shown to inhibit CIA development by reversing arthritis and blocking immune cells from accessing synovial tissue [169,170]. Importantly, VIB4920, unlike anti-CD40L monoclonal antibodies, inhibited B-cell activation but did not induce platelet aggregation in vitro, and no thrombotic side effects were encountered in any of the clinical studies [170]. Administration of an anti-CD40 antisense oligonucleotide using an amphoteric liposome-mediated delivery system to target macrophages and DCs also alleviates collagen-induced arthritis in mice. Amphoteric liposomes represent a novel vehicle concept for systemic and cell-directed antigen-presenting oligonucleotide delivery, which is a strong alternative to monoclonal antibody approaches to disrupt CD40–CD40L interactions [171]. A new generation of vectors called lipoplexes enables the delivery of small interfering RNAs (siRNAs) specific to a variety of molecular targets. The research focused on a specific cytosolic phospholipase A2 (cPLA2α) that inhibits the cascade involved in the production of prostaglandins in monocytes/macrophages [172]. In published experiments, cPLA2α siRNA distribution was associated with decreased expression and activity of cPLA2α in monocytes/macrophages, and pro-inflammatory cytokines such as TNF-α and IFN-γ were attenuated. Moreover, cell infiltration into joints of mice treated with cPLA2α lipoplexes was inhibited [173].

A new approach to macrophage targeted therapy is to change the macrophage phenotype by nanoparticle-mediated delivery of a plasmid encoding IL-10 to macrophages of rats with arthritis. This repolarizes the macrophage phenotype and prevents the progression of inflammation and joint damage [138]. Knowing that clodronate causes macrophage depletion, the effect of macrophage depletion with clodronate-containing liposomes (Clophosome) on the initiation and maintenance of RA was investigated. The study found that the degree of arthritis in the experimental group of mice was significantly reduced due to attenuation in infiltration of inflammatory cells in the joint cavity. In addition, the percentage of CD68^+^ macrophage cells in synovial cells was significantly lower than in the control group [7].

#### Tolerogenic Dendritic Cells (tolDCs) Induction

Due to the fact that the DC and monocyte lines are derived from a common progenitor, another new treatment method is worth mentioning. This method is tolDC-based immunotherapy, which is one of the most attractive approaches among innovative treatments for RA [16]. The main therapeutic approach is to differentiate a patient’s precursor cells (such as monocytes or bone marrow-derived stem cells) ex vivo into tolDCs, which are then loaded with the appropriate autoantigens and re-administered to the patient to restore tolerance to specific autoantigens of the T cells [174]. One of the more important studies was the study where tolDC was generated by treating human monocyte-derived DCs with NF-kB signaling inhibitor BAY 11-7082. TolDC, pulsed with four citrullinated peptide antigens, was named “Rheumavax” and a Phase I clinical trial was performed in Australia. Reumavax was well tolerated by patients with early RA and revealed promising data on efficacy in reducing the effector T cell population and increasing Treg cells [175]. Another very important Phase I study was “AutoDeCRA” by scientists in Newcastle. DC derived from human monocytes were loaded with autologous synovial fluid as a source of autoantigens and then patients received tolDC arthroscopically up to the knee with inflammation. The results of the study showed that intraarticular therapy with tolDC appears to be safe, possible and acceptable. Knee symptoms stabilized in two patients who received tolDC but no clinical systemic or immunomodulatory effects were detected [176]. Another study showed that DCs derived from monocytes of RA patients after modulation with dexamethasone and activated with monophosphoryl lipid A have the ability to develop tolerance traits at the transcriptional and translational levels. Additionally, the ability of tolDC to attenuate T cell responses to synovial antigens supports their potential for treating RA [177]. However, for the therapy to be fully effective, more research and answers to many important questions are needed. Thanks to numerous studies and new therapies, we are getting closer to precision medicine tailored to the needs of patients with RA. Undoubtedly, it is necessary to continue searching for new treatment options which are more cell-specific in order to make the efficacy and safety even better.

## 6. Application of Single-Cell Technologies in RA Monocytes/Macrophages Phenotyping

Uncovering the true diversity of cells is crucial for better understanding the functional heterogeneity of monocytes/macrophages in RA. Latest studies applying scRNA-seq have provided deeper insight into identifying new subsets of these cells [178]. It has been shown that by using scRNA-seq methods, monocytes are heterogeneous in the population of patients with active RA, revealing successive divisions. Transcriptional heterogeneity of synovial monocytes was driven by inflammatory cytokines and interferons, suggesting an important role of the local microenvironment on monocyte differentiation [179]. Other studies identified a novel subpopulation of anti-inflammatory MLS called MerTK^+^CD206^+^ synovial macrophages. These MerTK^+^CD206^+^ MLS treated with tyrosine-protein kinase (MerTK) agonists (UNC106258) were able to restore synovial homeostasis through pro-inflammatory S10012A reduction and induction of repair response of FLS in vitro [180]. On the contrary, using scRNA-seq, a subset of pro-inflammatory HBEGF^+^ macrophages were found. These cells were FLS and TNF-α-dependent. Indeed, HBEGF^+^ macrophages promoted fibroblast invasiveness and contributed to aggravating arthritis. Ex vivo studies of synovial tissue confirmed that most drugs used in the treatment of RA patients targeted HBEGF+ inflammatory macrophages [181]. However, further studies are expected to confirm that distinct subsets of monocytes/macrophages can influence the outcome of inflammation and ultimately may be useful in future personalized RA therapy.

## 7. Conclusions

Monocytes and macrophages are very important cells of the immune system that are involved in both initiating and resolving inflammation. They are crucial in RA, they produce cytokines that increase inflammation and contribute to the destruction of cartilage and bone. In addition to their paracrine role, they also show autocrine activity, causing further feedback loops, maintaining their own survival and activation, and escalating the inflammatory environment. Numerous studies have shown a high degree of heterogeneity among monocytes and macrophages in RA. This diversity is influenced by differences in origin, function, potential polarization and the influence of genetic, epigenetic and environmental factors. New diagnostic and therapeutic approaches are being developed with an increased understanding of the function and interaction between cells of the immune system. These new diagnostic and therapeutic methods for tuning of macrophages and monocytes presented in this review are an important component of precision medicine approaches to the treatment of RA. With the drugs listed here and with further research, achieving a strategy tailored to individual needs is becoming more and more achievable.

## Figures and Tables

**Figure 1 cells-10-01860-f001:**
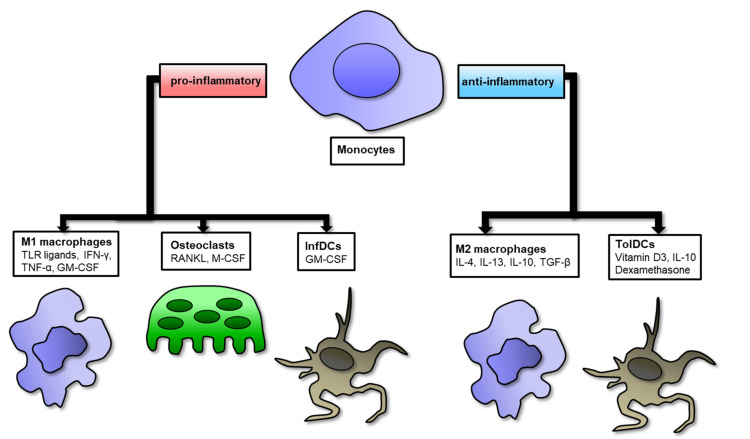
Schematic representation of macrophage polarization and its various stimuli differentiating toward pro- or anti-inflammatory phenotypes in RA. On the left side, monocytes can differentiate toward pro-inflammatory M1 macrophages, osteoclasts and inflammatory DCs (InfDCs); on the right side, monocytes can differentiate toward anti-inflammatory M2 macrophages and tolerogenic DCs (TolDCs).

**Figure 2 cells-10-01860-f002:**
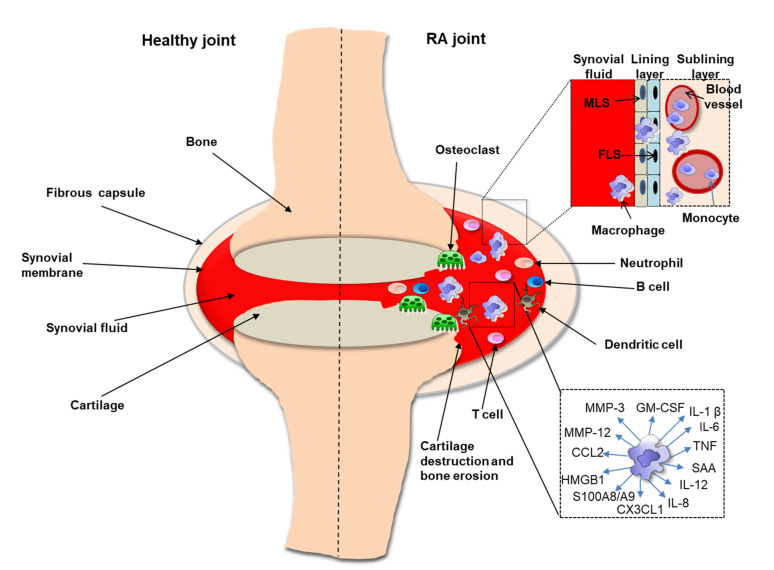
Schematic representation of differences between the RA joint (right) and a healthy joint (left). Monocytes/macrophages are the main producers of the pro-inflammatory cytokines that cause arthritis, cartilage destruction and bone loss in the RA joint. Involved in arthritis are also neutrophils, B cells, T cells, dendritic cells and osteoclasts. The synovial membrane is formed by a lining layer and a sublining layer. The lining layer it is made up of MLS (macrophage-like synovial cells) and FLS (fibroblast cells). Non-resident macrophages that arise from monocytes enter the joint from blood vessels.

**Figure 3 cells-10-01860-f003:**
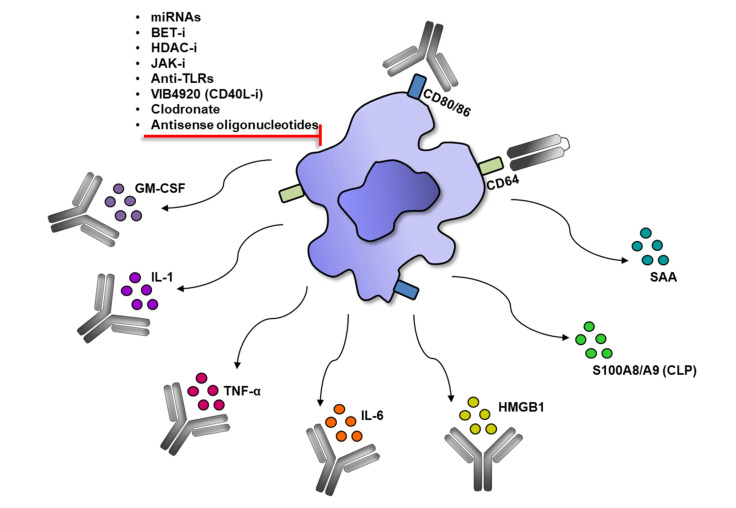
Modulation of macrophage in RA. Activated macrophages produce pro-inflammatory molecules which can be neutralized by monoclonal antibodies or inhibited by chemical agents and oligonucleotides.

**Table 1 cells-10-01860-t001:** Therapeutic drugs modulating monocytes/macrophages in RA.

Drugs	Immune Target	Clinical Stage	Main Course of Action on Macrophages/Monocytes
Infliximab	TNF	Approved	Neutralization of TNF, apoptosis of monocytes and macrophages in both peripheral blood and synovial fluid.
Adalimumab		Approved	
Etanercept		Approved	
Golimumab		Approved	
Certolizumab pegol		Approved	
Tocilizumab	IL-6	Approved	Inhibition of IL-6, apoptosis of monocytes.
Sarilumab		Approved	
Olokizumab		Phase III	
Clazakizumab		Phase IIb	
Vobarilizumab		Phase III	
Sirukumab		Phase III	
Anakinra	IL-1	Approved	Inhibition of IL-1.
Canakinumab		Phase II	
Rilonacept		Phase II	
Mavrilimumab	GM-CSF	Phase IIb	Blocking of GM-CSF, inhibition of M1 polarization.
Gimsilumab		Phase I	
Otilimab		Phase III	
Namilumab		Phase II	
Lenzilumab		Phase II	
Tofacitinib	JAK1/JAK3	Approved	Blocking of JAK-STAT pathway activity and macrophage-secreted inflammatory factors.
Baricitinib	JAK1/JAK2	Approved	
Upadacitinib	JAK1	Approved	
Spebrutinib	BTK	Phase II	Blocking of macrophage-secreted TNF-α, nitric oxide and IL-Iβ.
Fenebrutinib		Phase II	
Abatacept	CD80/CD86	Approved	Prevention of co-stimulation between APCs and T cells. Inhibition of monocyte differentiation into osteoclasts.
Ustekinumab	IL-12/IL-23	Phase II	Inhibition of IL-12/IL-23.
Guselkumab	IL-23	Phase II	Inhibition of IL-23.
H22 (scFv) -MAP	CD64	Preclinical	CD64+ M1 macrophage apoptosis.
VIB4920	CD40L	Phase II	Blocking the access of macrophages to the synovial tissue.
Clodronate-containing liposomes	Release chlorophosphate	Preclinical	Macrophage depletion.
NI-0101	TLR-4	Phase II	Blocking innate inflammatory responses by blocking TLR-4 activation.
TAK-242	TLR-4	Preclinical	Blocking innate inflammatory responses by blocking TLR-4 activation.

## Data Availability

Not applicable.

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
