# Peer review of "Tuning Monocytes and Macrophages for Personalized Therapy and Diagnostic Challenge in Rheumatoid Arthritis"

_cells, 2021, doi:10.3390/cells10081860_

Round 1

Reviewer 1 Report

 This is a comprehensive review regarding the role and therapeutic potential of targeting macrophage biology in rheumatoid arthritis. A few comments to improve the clarity and  the presentation of the material.

The abstract should follow the order of the article and the last two sentences should be removed. There should be text added to the early part of the article to reflect that epigenetic data and biomarkers for disease activity. The abstract should be weighted by the material presented.

These sentences are difficult to understand and should be removed from the abstract.

“We have also demonstrated potential consequences of epigenetic modification in cell-mediated inflammation. In addition, we have highlighted unique monocytes/macrophages signatures which contribute to identification of new biomarkers of disease activity and predicting treatment response in RA.”

CD14 ++ CD16- may be better as CD14hiCD16- rather then using two ++

line 138 “Based on GWAS study, “

There are two citations so is this based on GWAS studies? please define GWAS.

line 149 Many research should read Many research studies

Define the acronyms (MTX, TNFi, NGS, Ssc, FLS TET1 etc. ) at time of the first use

line 262 cytokines production should be cytokine production

Figure 3. Schematic representation of therapeutic modulation of macrophage in RA: should be modulation of macrophages in RA

neutralised

Delete the last line of the figure legend as it is not a therapeutic modulation but a different topic.

Line 361 This important role of …

line 400: EULAR and ACR recommendations

Author Response

Reviewer no. 1

This is a comprehensive review regarding the role and therapeutic potential of targeting macrophage biology in rheumatoid arthritis. A few comments to improve the clarity and the presentation of the material.

The abstract should follow the order of the article and the last two sentences should be removed. There should be text added to the early part of the article to reflect that epigenetic data and biomarkers for disease activity. The abstract should be weighted by the material presented.

These sentences are difficult to understand and should be removed from the abstract.

“We have also demonstrated potential consequences of epigenetic modification in cell-mediated inflammation. In addition, we have highlighted unique monocytes/macrophages signatures which contribute to identification of new biomarkers of disease activity and predicting treatment response in RA.”

Thank you for these comments and suggestions. The abstract has been re-arranged and reflects the material presented in the manuscript in the correct order. Also the last sentences have been removed.

CD14 ++ CD16- may be better as CD14hiCD16- rather then using two ++

This has been changed.

line 138 “Based on GWAS study, “

There are two citations so is this based on GWAS studies? please define GWAS.

This has been changed.

line 149 Many research should read Many research studies

This has been amended.

Define the acronyms (MTX, TNFi, NGS, Ssc, FLS TET1 etc. ) at time of the first use

These acronyms have been defined.

line 262 cytokines production should be cytokine production

This has been corrected.

Figure 3. Schematic representation of therapeutic modulation of macrophage in RA: should be modulation of macrophages in RA

This has been corrected.

neutralised

This has been corrected.

Delete the last line of the figure legend as it is not a therapeutic modulation but a different topic.

The last line has been deleted. 

Line 361 This important role of …

This has been amended.

line 400: EULAR and ACR recommendations

This has been amended.

Reviewer 2 Report

In this review, Roszkowski and Ciechomska discuss strategies aiming at modulating macrophage phenotype to mitigate inflammation and bone erosion in arthritis. The review is comprehensive and highlights various therapeutic approaches that have been investigated and developed including the use of biotherapies, small molecules and RNA interference.

Please find below my main concerns

-Perhaps the section dedicated to the “5.3. Modulation of monocytes/macrophages by other therapeutic modalities” could be structured differently i.e divided into distincts and more developed sections. Indeed, it contains the use of cytokines / fusion proteins, monoclonal antibodies and antisense/ RNAi therapeutic approaches in preclinical models of arthritis. For instance, the authors mention a study targeting cPLA2 but many other targets (cytokines , kinases ) have been investigated using various delivery strategies/vehicles.

-Because the manuscript is intended to address diagnostic challenges in RA and macrophage tuning for personalized medicine, it would profit from more specific insights into the recently described heterogeneity (using single cell RNA sequencing) of synovial macrophages in a final section. Distinct macrophage subsets may impact the outcome of inflammation and therapies, and deciphering their respective roles and functions might serve for future personalized therapy in RA.

I would suggest among others the following seminal articles/reviews that could be cited in this section:

Fan Zhang et al. “Defining inflammatory cell states in rheumatoid arthritis joint synovial tissues by integrating single-cell transcriptomics and mass cytometry”

Stefano Alivernini et al. “Distinct synovial tissue macrophage subsets regulate inflammation and remission in rheumatoid arthritis”

David Kuo et al. “HBEGF+ macrophages in rheumatoid arthritis induce fibroblast invasiveness”

Nicole Hannemann et al. “New insights into macrophage heterogeneity in rheumatoid arthritis”

Author Response

Reviewer no. 2

Comments and Suggestions for Authors

In this review, Roszkowski and Ciechomska discuss strategies aiming at modulating macrophage phenotype to mitigate inflammation and bone erosion in arthritis. The review is comprehensive and highlights various therapeutic approaches that have been investigated and developed including the use of biotherapies, small molecules and RNA interference.

Please find below my main concerns

-Perhaps the section dedicated to the “5.3. Modulation of monocytes/macrophages by other therapeutic modalities” could be structured differently i.e divided into distincts and more developed sections. Indeed, it contains the use of cytokines / fusion proteins, monoclonal antibodies and antisense/ RNAi therapeutic approaches in preclinical models of arthritis. For instance, the authors mention a study targeting cPLA2 but many other targets (cytokines , kinases ) have been investigated using various delivery strategies/vehicles.

Thank you for these comments and suggestions, however we decided to keep the structure as previously because some sub-sections ie. fusion proteins might contain only two sentences. Only tolDCs sub-section has been separated due to substantial amount of research studies which have been undertaken/published regarding tolDCs. 

-Because the manuscript is intended to address diagnostic challenges in RA and macrophage tuning for personalized medicine, it would profit from more specific insights into the recently described heterogeneity (using single cell RNA sequencing) of synovial macrophages in a final section. Distinct macrophage subsets may impact the outcome of inflammation and therapies, and deciphering their respective roles and functions might serve for future personalized therapy in RA.

Additional chapter 6 has been provided. This chapter is focusing on application of scRNA-seq technologies in identification of monocytes/macrophages heterogeneity in RA.

I would suggest among others the following seminal articles/reviews that could be cited in this section:

Fan Zhang et al. “Defining inflammatory cell states in rheumatoid arthritis joint synovial tissues by integrating single-cell transcriptomics and mass cytometry”

Stefano Alivernini et al. “Distinct synovial tissue macrophage subsets regulate inflammation and remission in rheumatoid arthritis”

David Kuo et al. “HBEGF+ macrophages in rheumatoid arthritis induce fibroblast invasiveness”

Nicole Hannemann et al. “New insights into macrophage heterogeneity in rheumatoid arthritis”

All these references have been cited in the manuscript: 180, 181, 182, 179 respectively.

Reviewer 3 Report

A comprehensive review of the therapeutic that are currently being used or are under development for the treatment of rheumatoid arthritis. Since a large part of the review focusses on monocytes, it would be good to include these cells in the title.

Lines 29-31: The statement in lines 29-31 is misleading, because the economic burden of Eur240 billion applies to all RMDs, of which osteoarthritis (OA) is the most common; even if RA is the most common inflammatory RA, it is far less common than OA. This statement should be rephrased to more accurately indicate what the economic burden of RA is.  I am also not sure that a Horizon 2020 report is the most appropriate reference – the original source should be referenced.

Lines 45-48 and throughout the manuscript: the authors should be more specific for each statement/finding whether it relates to human (in vivo or in vitro) or in vivo animal studies.

Line 49-50 and throughout manuscript: the M1/M2 macrophage classification is outdated and is largely based on in vitro data; recent seminal papers have defined synovial macrophage subsets through single cell sequencing – these papers should be referenced and this updated classification of macrophages should described: Zhang F et al. Defining Inflammatory Cell States in Rheumatoid Arthritis Joint Synovial Tissues by Integrating Single-Cell Transcriptomics and Mass Cytometry. Nat Immunol (2019) 20(7):928–42. doi: 10.1038/s41590-019-0378-1; Alivernini S et al. Distinct Synovial Tissue Macrophage Subsets Regulate Inflammation and Remission in Rheumatoid Arthritis. Nat Med (2020) 26:1295–306. doi: 10.1038/s41591-020-0939-8

Line 72: whilst leaving protective immunity. – should be: whilst leaving protective immunity intact. Also, a reference is required here (ref 174 I assume). In addition, Dex/VitD3 is only one example by which monocytes can differentiate into tolDC – other factors include for example IL-10.

Figure 2: another option is that monocytes differentiate into CD1c+ inflammatory DC under the influence of GM-CSF – for example see Reynolds G et al. Synovial CD4+ T-cell-derived GM-CSF supports the differentiation of an inflammatory dendritic cell population in rheumatoid arthritis. Ann Rheum Dis. 2016 May;75(5):899-907. doi: 10.1136/annrheumdis-2014-206578.

Line 88: subgroup – should be subgroups

Line 146 – patters – should be patterns

Line 224: do the authors mean predicated – or predicted?

Line 400: ACER – do the authors mean ACR?

Line 421: producing should be produced

Line 569 – ref 179: Authors please check this reference: from the title it sounds like that etanercept blocked migration of DC to the draining lymph node – not tolDC.

Table 1: Rheumvax: seems out of place in this Table.  It is a tolerogenic DC treatment and is not a ‘drug’ that modulates (or is designed to modulate) monocytes/macrophages in RA. 

Line 564/565: this statement seems to be out of place, given that two clinical trials with tolDC have already been mentioned.

Author Response

Reviewer no. 3

A comprehensive review of the therapeutic that are currently being used or are under development for the treatment of rheumatoid arthritis. Since a large part of the review focusses on monocytes, it would be good to include these cells in the title.

Thank you for these comments and suggestions. The title has been amended.

Lines 29-31: The statement in lines 29-31 is misleading, because the economic burden of Eur240 billion applies to all RMDs, of which osteoarthritis (OA) is the most common; even if RA is the most common inflammatory RA, it is far less common than OA. This statement should be rephrased to more accurately indicate what the economic burden of RA is.  I am also not sure that a Horizon 2020 report is the most appropriate reference – the original source should be referenced.

These sentences have been corrected.

Lines 45-48 and throughout the manuscript: the authors should be more specific for each statement/finding whether it relates to human (in vivo or in vitro) or in vivo animal studies.

In vivo or in vitro studies on human or animal models have been added.

Line 49-50 and throughout manuscript: the M1/M2 macrophage classification is outdated and is largely based on in vitro data; recent seminal papers have defined synovial macrophage subsets through single cell sequencing – these papers should be referenced and this updated classification of macrophages should described: Zhang F et al. Defining Inflammatory Cell States in Rheumatoid Arthritis Joint Synovial Tissues by Integrating Single-Cell Transcriptomics and Mass Cytometry. Nat Immunol (2019) 20(7):928–42. doi: 10.1038/s41590-019-0378-1; Alivernini S et al. Distinct Synovial Tissue Macrophage Subsets Regulate Inflammation and Remission in Rheumatoid Arthritis. Nat Med (2020) 26:1295–306. doi: 10.1038/s41591-020-0939-8

New classification has been updated and mentioned in the Introduction (line 61-63) and in Chapter 6. Additional references have been included in the manuscript: 180, 181 respectively.

Line 72: whilst leaving protective immunity. – should be: whilst leaving protective immunity intact. Also, a reference is required here (ref 174 I assume). In addition, Dex/VitD3 is only one example by which monocytes can differentiate into tolDC – other factors include for example IL-10.

These have been corrected.

Figure 2: another option is that monocytes differentiate into CD1c+ inflammatory DC under the influence of GM-CSF – for example see Reynolds G et al. Synovial CD4+ T-cell-derived GM-CSF supports the differentiation of an inflammatory dendritic cell population in rheumatoid arthritis. Ann Rheum Dis. 2016 May;75(5):899-907. doi: 10.1136/annrheumdis-2014-206578.

Additional sentences have been included (line 69-72) and Figure 1 has been amended accordingly.

Line 88: subgroup – should be subgroups

This has been corrected.

Line 146 – patters – should be patterns

This has been corrected.

Line 224: do the authors mean predicated – or predicted?

This has been corrected.

Line 400: ACER – do the authors mean ACR?

This has been corrected.

Line 421: producing should be produced

This has been corrected.

Line 569 – ref 179: Authors please check this reference: from the title it sounds like that etanercept blocked migration of DC to the draining lymph node – not tolDC.

This has been corrected and removed.

Table 1: Rheumvax: seems out of place in this Table.  It is a tolerogenic DC treatment and is not a ‘drug’ that modulates (or is designed to modulate) monocytes/macrophages in RA.

Rheumvax has been removed from the table.

Line 564/565: this statement seems to be out of place, given that two clinical trials with tolDC have already been mentioned.

These sentences have been removed.

Round 2

Reviewer 2 Report

The authors have addressed the reviewers' concerns and suggestions. No further comments.